# Artificial Intelligence: The Attitude of the Public and Representatives of Various Industries

**Tatjana Vasiljeva \*, Ilmars Kreituss and Ilze Lulle**

Faculty of Business and Economics, RISEBA University of Applied Sciences, LV-1048 Riga, Latvia;
ilmars.kreituss@riseba.lv (I.K.); ilze.lulle@gmail.com (I.L.)
\* Correspondence: tatjana.vasiljeva@riseba.lv; Tel.: +371-29487303

**Abstract:** This paper looks at public and business attitudes towards artificial intelligence, examining the main factors that influence them. The conceptual model is based on the technology–organization–environment (TOE) framework and was tested through analysis of qualitative and quantitative data. Primary data were collected by a public survey with a questionnaire specially developed for the study and by semi-structured interviews with experts in the artificial intelligence field and management representatives from various companies. This study aims to evaluate the current attitudes of the public and employees of various industries towards AI and investigate the factors that affect them. It was discovered that attitude towards AI differs significantly among industries. There is a significant difference in attitude towards AI between employees at organizations with already implemented AI solutions and employees at organizations with no intention to implement them in the near future. The three main factors which have an impact on AI adoption in an organization are top management's attitude, competition and regulations. After determining the main factors that influence the attitudes of society and companies towards artificial intelligence, recommendations are provided for reducing various negative factors. The authors develop a proposition that justifies the activities needed for successful adoption of innovative technologies.

**Keywords:** artificial intelligence (AI); public attitude; industry representatives' attitude; TOE framework; innovative technology; ICT industry

## 1. Introduction

Nowadays businesses need to gradually adopt new technologies such as social networks, 3D printers, robotics, artificial intelligence and big data to increase their performance, reduce costs and therefore be competitive in the market environment. Even public institutions have been introducing the emerging technologies to obtain higher performance and follow the latest information and communications technology (ICT) trends, adopting modern applications and benefiting from them.

Technological innovation has led to the digitalization era in business. Cloud computing, the Internet of things (IoT), big data, data science, artificial intelligence (AI) and blockchain are the rising technologies that have impacted the way business operates in recent years (Tekic et al. 2019; Johannessen 2020; Reez 2021; Saksonova and Kuzmina-Merlino 2019). Organizations across the world and even small and medium enterprises are paying more attention to the development and introduction of innovative technologies in order to adequately adapt their strategies to new market needs and stimuli. Digitalization trends and competitive pressure have changed the way business is operated and managed (Manesh et al. 2021; Saksonova and Papiashvili 2019).

All industries face numerous challenges during the process of implementing new technology. One such challenge is societal and organizational attitudes towards the technology. The first thing people usually think of in terms of AI is robots, and due to articles in magazines they are perceived as human-like machines that wreak havoc on Earth (Frankenfield 2021).

Bughin et al. (2018) believe that negative externalities may be sufficiently large if the socioeconomical system is not taken into account while implementing AI systems. The authors provide an example where the old business model needs to be changed and extensive job relocation has to be carried out due to adoption of AI. In the given case, negative externalities may be sufficiently large and create the risk of a societal backlash against AI that could limit the full potential anticipated from these technologies.

Given these trends, companies should take care of their teams and try to reduce and mitigate employees' resistance to and fears of coming changes, raising the firms' readiness for future challenges. Negative employee attitude towards ICT trends has been the object of research for many scholars (Bean 2019; Caputo et al. 2019; Lichtenthaler 2020a; Zaki 2019; Vasiljeva and Novinkina 2019), who have paid special attention to factors influencing the attitude towards technological changes.

The objective of this study is to explore the factors which influence society's attitude towards new ICT trends and the adoption of innovative technologies in business. The paper argues that considering all impacting factors and following the best practice, society and entrepreneurship could be the main predictors and prerequisites for successful business growth and the evolution of society.

Currently there is a lack of scientific research on public and organizational acceptance or attitudes towards AI, which opens a wide range of opportunities to investigate the factors affecting AI implementation in many areas (Davenport and Mahidhar 2018; Pumplun et al. 2019; Pillai and Sivathanu 2020). However, Kaplan and Haenlein (2019a, 2019b) emphasize that AI is still in its infancy and it is difficult to predict how it will develop in the near future. For a better understanding and implementation of AI, the world must consider AI requirements for enforcement, employment, ethics, education, entente and evolution.

Many researchers believe that "employees' attitude has a key role in the adoption of new technologies and may strongly affect technology acceptance decisions" (Lichtenthaler 2020b). Numerous studies are devoted to the investigation of such "non-tangible resources" as social networks, virtual realities, artificial intelligence, etc. (Caputo et al. 2019; Borges et al. 2020; Gursoy et al. 2019; Sujata et al. 2019; Holmlund et al. 2020). At the same time, companies' human resources and their attitude and perception towards the adoption of new technologies should be considered as a crucial factor of business competitiveness and success. Only a few studies have investigated the digital transformation and technological evolution of companies from the perspective of personnel's attitude towards technological changes. This study tries partly to close this gap by examining the attitude of the public and representatives of various industries towards the adoption of artificial intelligence.

The originality of the paper lies in the research of the artificial intelligence phenomenon in the context of industries' and society's attitude and the reinvention of the main business processes at an enterprise. Therefore, this paper contributes to the topical discussion on business strategy for digital transformation and the usage of artificial intelligence to raise the competitiveness of a company. Namely, the paper demonstrates the business and public attitudes towards AI and what may trigger the success and acceptance of AI in society.

The objective of the study was to determine the main factors that influence the attitude of society and companies towards artificial intelligence and provide recommendations for reducing various negative factors aimed at entrepreneurship.

More than 400 respondents and 13 experts from Latvia participated in the research process during spring 2021.

Latvia is very similar to the other Baltic countries in terms of the level of employees' education, societal mentality and readiness for coming changes. Thus, the authors believe that the perception of the latest ICT trends is also similar and that the results can be generalized to some extent.

The authors of the present study selected enterprises for investigation which belong to local business and to international business and are situated in Latvia. Firstly, the aim was to gauge their readiness for AI introduction and implementation in business processes. Secondly, we were interested in clarifying to what extent public attitude towards

AI implementation is ill-informed and uncertain due to insufficient knowledge and mistrust of AI systems, which can slow down the development of AI systems in business processes. This paper shows originality by analyzing the public and business attitudes towards broad AI implementation in social life.

The paper proceeds as follows. The first part is devoted to the substantiation of the theoretical framework of the research, and the second part shows the results of the empirical research. The paper finishes with a discussion on managerial contribution, research limitations and further research areas.

## 2. Theoretical Background

### 2.1. Literature Analysis

Examining the impact of AI might be essential for every industry, as finance, manufacturing, retail, health care, etc., are all potentially disrupted by the onslaught of AI, robotics and similar innovative technologies. Digital technology improvements allow companies to optimize their own processes, for example, through inventory management and efficient material processing, and they enable companies to transform supplier and customer interfaces, for example, through increased customer knowledge and material tracking (Ranta et al. 2021; Borges et al. 2020; Caputo et al. 2019). Additionally, entrepreneurs utilize digital technologies to develop blended value propositions, which merge environmental, social and financial value for their stakeholders (Gregori and Holzmann 2020).

Digital trends and technologies have led to the era of Industry 4.0. This has opened discussions about advantages relevant to resource and labor productivity, radical changes to business models, and social challenges (Mavropoulos and Nilsen 2020), especially those pertaining to circular economy concepts (Pagoropoulos et al. 2017).

Reckwitz (2020) claims that the typically modern logic of rationalization, formalization and standardization is being replaced by the logic of the special, the extraordinary and the original. This means that people have become personalities, with technology playing a central role in fostering this new era. It is only the omnipresence of quantification, measurement and comparison (in the form of ratings, rankings and scores) that ultimately contributes to the predominance of the special. This is all supported by the exponentially growing possibilities of data production and processing in the digital age.

If humankind is to benefit from the new technological developments, they will need to develop new skills and competencies. For example, they have to develop critical competencies, which are based on the knowledge needed to enable reflection on the ethical and value-related challenges of the new technological developments (Johannessen 2020). They also include the ability to recognize patterns and understand what is happening to people, organizations, society, the environment and the planet (Lima et al. 2017).

Other human competencies in demand will relate to strong and enlightened leadership. Leaders will be needed who understand the approaching era, rather than leaders with a narrow vision and knowledge based on the previous industrial era (Johannessen 2020).

Many researchers have focused on the issue of developing new business strategies towards digital transformation and embedding AI in various business processes such as value chains (Kitsios and Kamariotou 2021; Borges et al. 2020; Oosthuizen et al. 2020). A strategic framework for customer experience management has become a challenge for many organizations (Holmlund et al. 2020). Some researchers truly believe AI tools can narrow the gap between businesses and clients, assist retail organizations to get better client involvement and significantly improve customer experience (Sujata et al. 2019).

Zaki (2019) reasonably considers digital transformation as the step of innovation that seriously impacts the workforce and their attitude towards coming changes. He maintains that "employees should keep an eye out for their next job, which in all likelihood will be a very different job".

Some authors even see some fear of introduction of AI as a partly disruptive innovation (Lichtenthaler 2020a, 2020b; Bean 2019) reflecting to what degree AI may replace and extend human intelligence. Ulrich Lichtenthaler (2020a, 2020b) is concerned to what extent

"employees are reluctant to adopt and accept the latest technological innovations and may even be frightened by them".

Despite the many research publications on various AI aspects and the challenges AI tools pose for business, there is still a research gap. We do not know to what extent our society is ready for the coming changes or what individuals' perceptions and attitudes are with regard to such innovations as artificial intelligence, various chatbots and other similar challenging tools. Closing this research gap could help researchers and businesses understand what the most negative factors are in terms of public resistance to AI acceptance in order to prevent or mitigate them.

Artificial intelligence is a branch of computer science. The basic concept of AI is to combine large amounts of data with fast processes and superior algorithms. In the end, this makes it possible for systems to learn from patterns without being explicitly programmed (Kaplan and Haenlein 2019b).

There is no single clear definition of AI, since this can vary based on AI's goal. As can be seen in Table 1, the definitions on the right measure against an ideal concept, which deals with helping machines find solutions to complex problems, whereas the ones on the left measure success in terms of human performance. Therefore, a tension exists between approaches centered on humans and approaches centered on rationality (Deshpande 2009).

**Table 1.** Definitions of artificial intelligence based on their approach. Adopted from Deshpande (2009).

| Think Like Humans | Think Rationally |
|---|---|
| "The exciting new effort to make computers think ... machines with minds, in the full and literal sense" (Haugeland 1989) "The automation of activities that we associate with human thinking, activities such as decision-making, problem solving, learning ..." (Bellman 1978) | "The study of mental faculties through the use of computational models" (Charniak and McDermott 1985) "The study of the computations that make it possible to perceive, reason, and act" (Winston 1992) |
| **Act Like Humans** | **Act Rationally** |
| "The art of creating machines that perform functions that require intelligence when performed by people" (Kurzweil 1990) "The study of how to make computers do things at which, at the moment, people are better" (Rich and Knight 1991) | "A field of study that seeks to explain and emulate intelligent behavior in terms of computational processes" (Schalkoff 1990) "The branch of computer science that is concerned with the automation of intelligent behavior" (Luger and Stubblefield 1993) |

Indriasari et al. (2019) have characterized predictive analytics as a form of AI application in financial institutions, which can be used in various areas to improve financial business effectiveness, risk management and prevention, operational efficiency, customer relationships and revenue growth. All these applications can be used in other industries as well; for example, AI has boosted marketing and customer relationship management (CRM) regardless of industry. AI can create a comprehensive profile of current or potential customers based on various types of data, such as volume and number of financial transactions, frequency and type of past operations, current web browsing behavior, psychographic and demographic characteristics and interactions with the company (Paschen et al. 2019). Indriasari et al. note that in financial service institutions, such fields as operational efficiency and financial efficiency have been investigated by academic research while other areas like customer relationship, risk prevention and business efficiency have not been investigated (Indriasari et al. 2019). This opens a wide space for various kinds of academic research which could have valuable practical implications.

Jöhnk et al. (2020) have proposed a framework for how to evaluate organizations' readiness for AI implementation. The model consists of 18 AI readiness factors grouped in five categories which organizations should evaluate before AI adoption: (1) strategic alignment; (2) resources; (3) knowledge; (4) culture; and (5) data.

AI system development encounters numerous points of contention and technical challenges (Amershi et al. 2019). Some of these problems can be resolved during the project's planning phase if business and IT divisions have a common understanding of the project (Takeuchi and Yamamoto 2020). Cubric (2020) has summarized the barriers for AI adoption in business and management, which include economic and technical aspects related to the prohibitive cost of implementation and maintenance, the need for support infrastructure, lack of usable data, non-reusability of models and limited applicability for some classes of problems. Equally important are social barriers such as lack of knowledge, safety and trust, dependence on non-humans, stakeholders' perspectives and excitement about jobs.

What can be done to overcome these economic and social barriers, obstacles and fears, and to what extent is society ready to accept AI in its various forms and representations?

### 2.2. Research Design and Methodology

The research question (RQ) addressed in this study is as follows:

**RQ**: What is the current public attitude towards AI? What is the current attitude of industries towards AI and do social factors hinder AI adoption?

This study aims to evaluate the current attitudes of the public and employees of various industries towards AI and investigate the factors that affect them. After extensive research of literature sources, we have found several factors which affect the attitudes of the public and business. In previous studies, researchers have used well-known frameworks to test the attitude and actual usage of new technologies, such as the technology acceptance model (TAM) or the technology–organization–environment framework (TOE).

The authors believe that the TOE framework, which has been used in other studies to test the factors affecting technology adoption, is relevant for this study. According to this approach, a company's decision to introduce a new technology is affected by technological, organizational and environmental factors (Baker 2012). Although the proposed model is more customary for organizational studies, we decided to extend the model and add social factors that influence the acceptance of AI in society. Descriptions of the variables relevant for the study are presented in Table 2.

**Table 2.** Variable descriptions.

| Variable | Explanation | Adapted From |
| --- | --- | --- |
| *Knowledge* | Due to the inherent complexity of AI, AI-based systems often lack transparency, which prevents people from adopting such technologies. In addition, people who lack knowledge have higher hopes for AI based on self-made assumptions and often these assumptions cannot be met. As a result, the person is dissatisfied with the integrated technology of AI (Brill et al. 2019). | Jöhnk et al. (2020) |
| *Trust* | Trust is perceived competence (i.e., credibility) and benevolence is the extent to which one feels secure and psychologically comfortable about depending on the trustee. In this context, there is a sense of trust that AI provides services as promised and accurately (Doney and Cannon 1997). | Wirtz et al. (2018) |
| *Cost effectiveness* | This variable indicates how the benefits of new technology adoption exceed the cost of such technology—to what extent the technology provides faster and more accurate results by saving time and costs (Pillai and Sivathanu 2020). | Pillai and Sivathanu (2020) |
| *Relative advantage* | The degree to which an innovation is perceived as being better than the idea it supersedes (Rogers Everett et al. 2019). | Pillai and Sivathanu (2020) |
| *Department readiness* | Individuals will not adopt a technology, though they might like it, if they lack money, resources and skills essential to adopt the technology (Pillai and Sivathanu 2020). | Pillai and Sivathanu (2020) |
| *Top management support* | Top management has a crucial role in the technology adoption process; they should explicitly and actively help the organization in the introduction process. Top management involvement and commitment acts as a motivator and helps to overcome employee resistance (Sony and Naik 2019). | Pillai and Sivathanu (2020) |

The conceptual model of the research includes six independent factors (trust, knowledge about AI, cost effectiveness, relative advantage, department readiness, top management support) that are classified in three dimensions (social, technological and organizational) and one dependent factor (attitude towards AI)—see Figure 1.

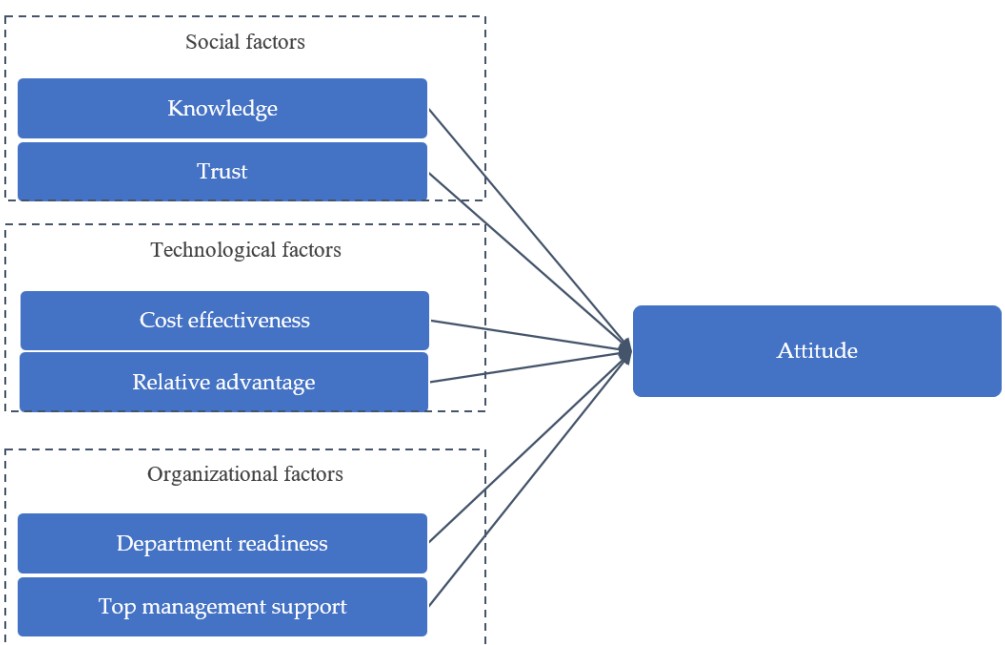

**Figure 1.** Conceptual model of the research (compiled using Pillai and Sivathanu (2020); Wirtz et al. (2018) and Jöhnk et al. (2020)).

## 3. Methodology and the Results of the Empirical Research

### 3.1. Research Time Frame and Data Collection

For the present research, we used the following methods to gather primary data: (a) a survey to measure all variables in the proposed conceptual model; and (b) semi-structured interviews to find out the reasonable opinion of professionals based on their experience.

3.1.1. Characteristics of Respondents

- The survey was conducted among those who have work experience and are at least 18 years old. The aim was to reach all social groups.
- Interviews were held with 2 groups of professionals. The first group was considered to be AI technological experts, who have a solid knowledge of AI technology, follow the latest AI trends, and are competent in business management. The second group comprised management representatives from different levels of management. The selection of representatives includes organizations of different sizes, markets, industries and levels of current AI adoption.

Design:

- The questionnaire was carried out through the website www.webropolsurveys.com, and the respondents were reached through social platforms or direct messaging via email. The survey consisted of 29 questions divided into 3 sections: demographic characteristics, organizational characteristics and questions to measure the 7 main factors of the study. All questions in the first and second sections were single-choice questions; 5-point Likert-type scale questions were used in the third section.
- All interviews took place online and the average time of an interview was 30 min. There were two sets of question, one for the AI experts and another for the management representatives. The questions were sent to the professionals prior to the

interview. The qualitative and quantitative data were gathered between December 2020 and March 2021.

A survey was conducted that incorporates all the constructs involved in the proposed model. Qualitative data were gathered from the interviews held with AI experts; top managers from various organizations of different industries, sizes and levels of current AI adoption; and finance industry representatives.

Three demographic variables (gender, age, department where the employee works) and four organizational variables (industry, size, business classification, current AI usage) were included in the survey questions to analyze differences of opinion among various social groups.

Quantitative data from the questionnaire were analyzed using statistical analysis methods. Firstly, all variables were tested as to whether they fit the proposed conceptual model and, secondly, correlation and regression analyses were done to determine the key predictors of attitude.

The survey was conducted among employed persons who were at least 18 years old; in total, 409 people responded to the survey completely (67% of all participants who had started responding).

### 3.1.2. Selection of Experts for the Interview

To compare and check the results of the survey, we additionally conducted semi-structured interviews with representatives of various industries. There were 13 interviews in total: 6 with AI experts, 7 with management representatives.

The criteria for selecting industry representatives as AI experts were as follows: (1) at least master's-level education (2) not less than 5 years' experience with new ICT technology; (3) more than 5 years' managerial experience. The criteria for selecting top managers as interviewees were similar, but the experience with managing a team engaged in implementing AI tools could be less—up to 3 years.

Most of the interviews were held through online platforms and the average time of an interview was 30 min. Qualitative data from the research interviews were analyzed, with a focus on determining the current situation in organizations regarding AI implementation in business processes, employee and employer attitudes towards this, and the main factors hindering AI adoption.

In order to answer the research questions, 2 models were used based on the proposed conceptual model. Model 1 tested all independent variables' impact on the dependent variable (knowledge, trust, cost effectiveness, relative advantage, top management support and department readiness) and Model 2 tested the grouped variable impact, referred to as social factors, technological factors and organizational factors, proposing a wider scope.

### 3.2. Analysis and Interpretation of Results

For primary data analysis, IBM SPSS software was applied. All variables were composed based on the mean value of all items representing the variable.

*Reliability and validity*: the high internal consistency of variables was confirmed, as the value of the Cronbach's alpha coefficient was above 0.6; as the Kolmogorov–Smirnov normality test showed that the data were not normally distributed, non-parametric tests were used in the further analysis (see Table 3). A generally accepted rule is that $\alpha$ of 0.6–0.7 indicates an acceptable level of reliability, and 0.8 or greater indicates a very good level. At the same time, a value for Cronbach's alpha (above 0.6), in our case for the variable "trust", was not very high. For all other variables, the meaning of Cronbach's alpha is in the interval 0.74–0.9, which could be interpreted as a good and reliable level. The authors still decided not to exclude any items in the variable "trust", assuming that the concept of trust is very intangible in nature and could be interpreted by the respondents in very different ways.

**Table 3.** Descriptive statistics for variables.

| Variable | N | N of Items | Mean | Std. Deviation | α | K-S test t-Value | K-S test Sig |
|---|---|---|---|---|---|---|---|
| *Knowledge* | 409 | 3 | 2.68 | 0.95 | 0.79 | 2.22 | <0.001 |
| *Trust* | 409 | 4 | 3.19 | 0.77 | 0.60 | 1.95 | 0.001 |
| *Cost effectiveness* | 384 | 3 | 3.40 | 1.00 | 0.89 | 2.75 | <0.001 |
| *Relative advantage* | 376 | 5 | 3.89 | 0.95 | 0.74 | 3.09 | <0.001 |
| *Department readiness* | 396 | 3 | 3.18 | 0.95 | 0.77 | 1.67 | 0.005 |
| *Top management support* | 338 | 3 | 3.33 | 1.00 | 0.87 | 2.20 | <0.001 |
| *Social factors* | 409 | 7 | 2.97 | 0.68 | 0.70 | 1.55 | 0.010 |
| *Technological factors* | 394 | 8 | 3.63 | 0.90 | 0.85 | 2.33 | <0.001 |
| *Organizational factors* | 393 | 6 | 3.24 | 0.91 | 0.89 | 1.34 | 0.041 |
| *Attitude* | 409 | 3 | 3.43 | 1.14 | 0.90 | 2.35 | <0.001 |

*Public attitudes*: distribution of the respondents according to their level of attitude is shown in Figure 2. We can see that 53% of respondents' attitudes towards AI were positive or very positive. The total attitude of the population is reflected as the average assessment from the survey data obtained. The measurements show that the average attitude towards AI in the population can be considered to a large extent as positive (M = 3.43, SD = 1.14).

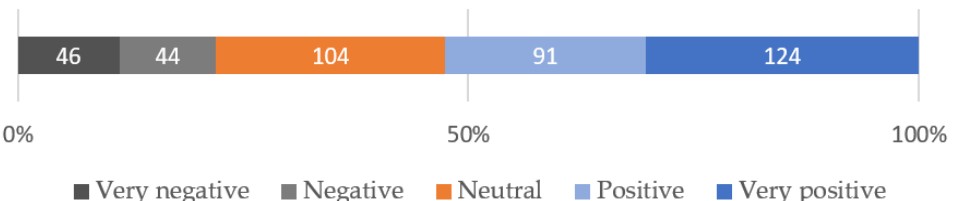

**Figure 2.** Respondent distribution by level of attitude.

It was observed that there were significant attitude differences between employees at companies with already implemented AI solutions (M = 4.26, SD = 0.81) and employees at companies not likely to implement any AI solution in the near future (M = 2.93, SD = 1.18), H(2) = 79.78, $p$ < 0.01. While only 35% of employees had a positive or very positive attitude in companies with no intention to implement AI solutions, more than twice as many employees had a positive or very positive attitude in companies with already implemented AI solutions, reaching 87% of all employees. Attitude level also differed among employees at companies of different sizes. The level of attitude for employees at micro and small companies was lower than for employees at medium-sized or large companies, H(3) = 25.58, $p$ < 0.01 (see Figure 3).

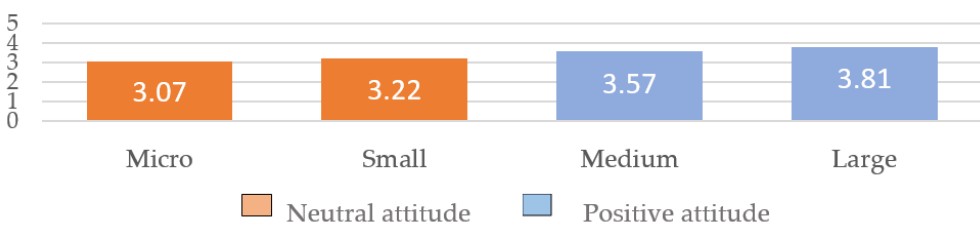

**Figure 3.** Attitude towards AI among employees according to company size.

Attitude differed among industries as well, H(12) = 27.62, $p$ = 0.006. As shown in Figure 4, the most positive attitudes were found in the ICT industry (M = 3.82, SD = 1.12),

followed by manufacturing (M = 3.73, SD = 1.05) and financial, legal and business services industries (M = 3.70, SD = 1.09). The least positive attitudes were found in the tourism, hospitality and entertainment industries, where the mean range was M = 2.70, SD = 1.30.

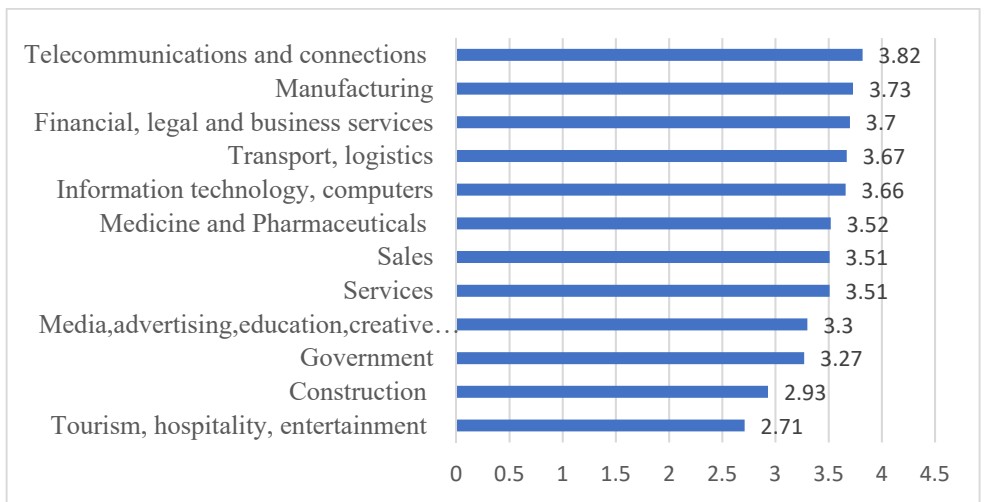

**Figure 4.** Average attitude towards AI according to industry.

A significant difference was observed in sales and financial, legal and business services industries within international companies (see Figure 5). Attitudes in the sales industry differed between domestic and international companies from M = 3.19, SD = 0.83, considered as neutral, to M = 3.86, SD = 1.07, considered as positive, while the financial, legal and business services industries reached an average attitude of up to M = 4.02, SD = 0.88 within international companies.

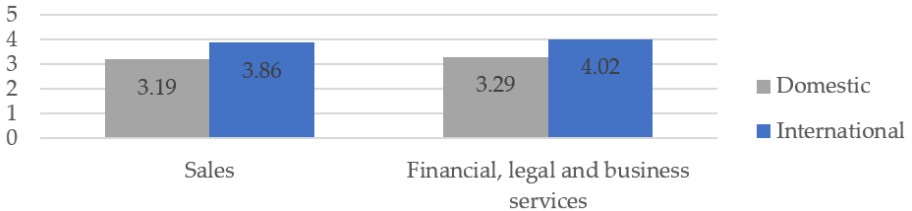

**Figure 5.** Sales and financial, legal and business services' level of attitude at international and domestic companies.

Public opinion of AI was discussed with the AI experts as well. They believed that most of society has overcome the negative attitude which was noticeable some years ago. Experts mentioned tenure as a factor which creates differences in attitudes. In conclusion, the average attitude towards AI in society was mainly positive, although it differed among various population groups.

*Main predictors*

To determine the main predictors of society's attitude towards AI, the authors analyzed the correlation and regression between the variables.

To test the predictors of attitude towards AI, two models were used:

- Model 1 for more comprehensive review of social factors, technological factors and organizational factors;
- Model 2 for more detailed influencing factor analysis was used for the analysis of knowledge, trust, cost effectiveness, relative advantage, department readiness and top management support.

Among all variables, the variable cost effectiveness had a strong relationship with attitude (rs(384) = 0.66, $p < 0.001$) followed by relative advantage (rs(376) = 0.55, $p < 0.001$). Relative advantage and cost effectiveness showed a high correlation as well (rs(366) = 0.60, $p < 0.001$), which makes all three variables interrelated (see Figure 6).

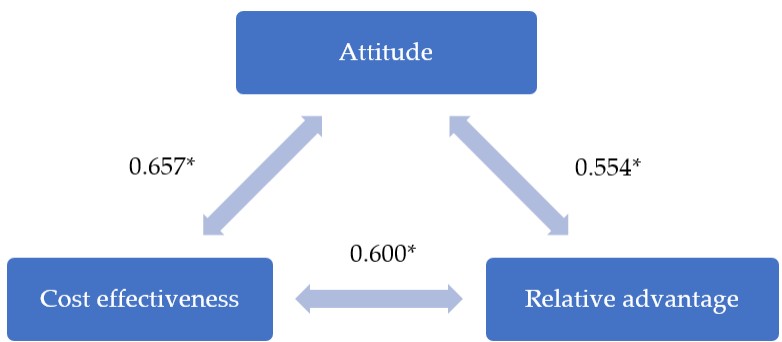

**Figure 6.** Relationship between the variables. * Correlation is significant at the 0.01 level (2-tailed).

As shown in Figure 7, cost effectiveness alone determined 78% of attitude level and knowledge determined 64% of the linear regression model. These observations within the industries show that there are different factors which affected the attitude towards AI. A high correlation with the social factors could lead to the need for explanations of how the system works to fully understand and trust the machine.

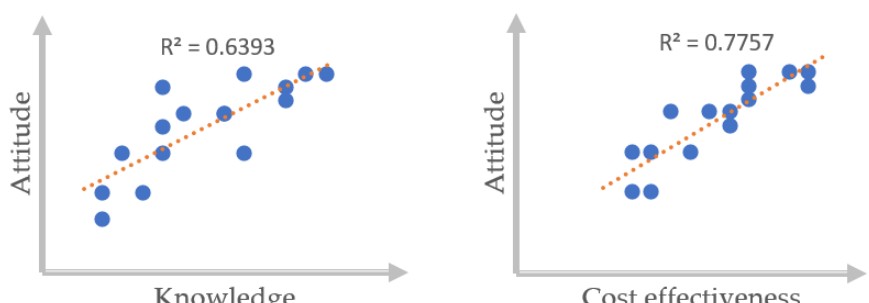

**Figure 7.** Significant correlations in the transport/logistics/transportation industry.

The regression analysis in Model 1 provided an adjusted $R^2$ equal to 0.556, meaning that the six independent variables explained 56% of the variance of the dependent variable attitude, F(6,310) = 64.72, $p < 0.001$, while 3 variables were found to have an insignificant impact on attitude: knowledge, department readiness and top management support. Hence, these factors were eliminated from Model 1 and the analysis was repeated.

As illustrated in Figure 8, the factor with the highest weight was cost effectiveness, followed by relative advantage and trust.

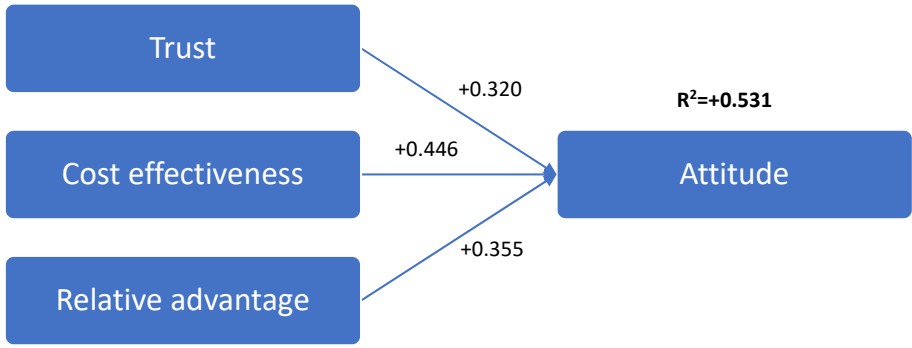

**Figure 8.** Regression model with coefficients for variables in Model 1.

As illustrated in Figure 9, the greatest influence on the value of attitude was exerted by technological factors and the equation to predict attitude is equal to 0.395 (social factor) + 0.785 (technological factor) − 0.587.

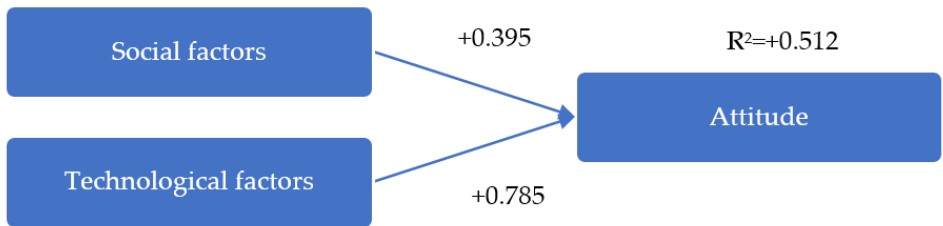

**Figure 9.** Regression model with coefficients for variables in Model 2.

*Business adoption*

Business adoption of AI is mainly discussed based on the expert interviews conducted for this research, as organizations' interest in digitalization and the latest technologies has increased in the past few years. To discuss the factors affecting AI adoption, we clustered the data based on the conceptual model proposed in this study. Based on the TOE model, we defined 3 groups of independent variables which affect AI adoption—technological factors, organizational factors and environmental factors—while discussions with the experts uncovered the great significance of the additional group business factors.

*Technological factors.* Technological readiness refers to the ability of the organization to adopt new technology; therefore, its observation offers a better way to foretell the benefits gained from technological implementation (AlSheibani et al. 2018). AI systems can lower production costs, save time and be more precise. The authors completely agree with other AI researchers that there are other factors predicting the actual implementation decision (Al-Maroof et al. 2020; Jarrahi 2018) and business factors are among the most important.

*Business factors.* Experts believed that most AI adoption problems start with insufficient investigation before starting the project. The authors agree with the opinion of experts from financial organizations that a project should not start with data, but with a business problem definition. The exact project scope must be defined and the type of tasks for automatization—routine or cognitive tasks—should be clearly defined.

Experts proposed the following sequence of questions, which should be answered by an organization before AI implementation:

- If there was an opportunity, what would you like to predict?
- How would this affect your business if you could?
- What data are needed to do this?

*Organizational factors.* This variable measures the availability of an organization's resources necessary for technology adoption. Expert opinion aligned with previous studies indicating that the top management has a significant impact on AI adoption (AlSheibani et al. 2018; Jöhnk et al. 2020). One expert claimed that "my job is to provide the business with information about the current technological trends in the industry, suitable AI application examples for the business and their benefits; further, it all depends on the manager".

Department readiness measures the availability of skills and resources needed for new technology adoption (Pillai and Sivathanu 2020) and employee readiness for IT was widely discussed among the experts, who confirmed the findings of previous studies that employees fear being laid off or becoming more regimented in the era of digitalization (Parry et al. 2016; Cubric 2020). The experts also believed that the attitude towards digitalization depended on management support as well as previous experience with automatized systems. It was confirmed that employees who have previous experience with automated solutions were more likely to accept AI implementation in their routine tasks. Managers should be the ones who encourage employees to adopt the new technology, reassure them

about job stability, explain their future role in the workplace, and encourage them to believe in the new technologies.

Experts mentioned the lack of broad and specialized knowledge of innovative software tools and business technology at IT departments as another factor influencing AI adoption. To overcome these influencing factors, experts suggested that universities should adapt their courses in both business administration and IT programs. To strengthen collaboration between IT and management, technological universities should include lectures about business operations and the importance of IT in them.

*Environmental factors.* One clear environmental factor is competition, which can have both a positive and a negative influence on AI adoption. Experts mentioned that competition is a very important factor not only for digital development, but development in all fields.

Experts believed that the COVID-19 pandemic has boosted the interest in AI applications. Many businesses had to rethink their strategy; as a result, a number of online businesses emerged. Virtual assistants, chatbots, robots, client targeting—all of these are AI applications.

Another important environmental factor is regulations. The EU developed a regulation regarding the use of AI technologies only in the year 2018, and there are a lot of misunderstandings regarding it. This creates confusion for AI vendors regarding the systems which they are implementing. The ethic codex is not very precisely developed as well, which slows down the implementation process. This creates confusion for businesses and can cause a negative attitude towards AI implementation. The unclear regulation also creates confusion for businesses and AI vendors about the data exchange process.

Although no other environmental factors were extracted from the interviews, it is necessary to emphasize the need for good vendor support. Selection of an AI vendor that matches the organization's needs is an essential element of the implementation process that can substantially foster the introduction of innovative technologies.

Additionally, experts clearly indicated the role of competition in business development as a factor influencing both the business directly and top management's attitude. This factor cannot be ignored by a company. If the current market situation shows that AI adoption is necessary, they must consider AI adoption. However, it is worth mentioning one expert's suggestion that competitors' announcements of their digital developments and innovations should be assessed with a balanced attitude. This can be linked to the previously discussed business factor—lack of clear specification of business needs.

The last factor the authors propose for discussion is collaboration between IT and business representatives. Such collaboration must be strengthened from both sides: IT should be more interested in business problems, while businesses should show more interest in gaining knowledge of innovative technologies so that they can be more competent in topics related to digitalization.

## 4. Discussion, Conclusions and Contribution of Research

The research led to the following conclusions.

Despite the fact that AI adoption in business and managerial processes started in the previous century, several factors still restrain the wide implementation of AI tools.

In order to have a deeper understanding of the main factors that influence the attitude of the public and companies, the three groups of factors determined by Tornatzky et al. (1990) and Baker (2012) in the TOE model (technological, organizational and environmental) were expanded and reinforced in this research. A new group—social factors—was added that includes such factors as knowledge and trust. Technological factors were supplemented with two new factors—cost efficiency and relative advantage. Organizational factors were expanded by top management support and department readiness.

The research showed that the average attitude towards AI in the population can be considered to a large extent as positive (53% of respondents' attitudes towards AI were positive or very positive, 26% neutral) and differed among various population groups.

Regression analysis showed that the most important groups of factors determining the attitude to AI were the technological, organizational and social groups. Looking at those groups in detail, the main factors with the highest weight that influenced the attitude of society towards artificial intelligence were found to be the technological factors cost effectiveness and relative advantage, the organizational factor top management support and the social factor trust. Three factors were found to have an insignificant impact on attitude: knowledge, department readiness and top management support.

AI experts had a similar opinion about the relationship between the technological factors, actual knowledge and attitude. They confirmed that most of society has overcome the negative attitude which was noticeable some years ago but still has a low level of knowledge regarding AI technologies. There is a part of society that has had interaction with AI-based projects or technologies, and they have an accurate view on the technological benefits of AI. Another part of society has been educated about AI principles, usage and benefits during university studies, workplace training or individual exploration of the topic. However, many people still have little idea what AI actually means, how much it affects our lives and decisions, and how much we use it in our daily activities.

The findings of the study clearly indicate the necessity of having appropriate skills, knowledge and qualified resources for an enterprise to be ready for innovative AI-based business processes and technological changes. Another social factor is trust of the accelerating technological changes and a positive perception of them. Caputo et al. (2019); Holmlund et al. (2020); and Lichtenthaler (2020a) have stressed the role of soft skills and the relevance of human resources in any innovative approach and especially digital transformation for achieving a breakthrough in business performance.

In order to leverage advanced technology and AI tools, companies should define their business requirements for the automatization of processes in a very clear way. Top managers have to acquire knowledge about the feelings, attitudes and motivations of their team to raise awareness of AI technology. Kitsios and Kamariotou (2021), Bean (2019), Cubric (2020) and Borges et al. (2020) have shown the importance of building AI into business and digital transformation, but there is still a potential for high ambiguity and uncertainty about people's attitudes towards new disruptive technological achievements. Underestimating employees' and the the public's attitudes towards the latest ICT tools, such as AI, blockchain technology, robotization, chatbots, etc., could have an extremely negative effect on any company. The overall public attitude towards AI can, to a great extent, be considered as positive, although it varied according to different organizational categories. It was revealed that employees working at international companies had a more positive attitude towards AI than employees from domestic companies, while there were no significant differences in opinion according to the individual's demographic measures, such as gender or age.

A statistically significant more positive employee attitude towards AI was found at international companies regardless of the industry, meaning that in general international experience with new technologies and AI plays a positive role. However, attitude towards AI differed significantly among industries. Representatives (employees) of the telecommunications and connections industry had the most positive attitude, followed by manufacturing and financial, legal and business services industries. The least positive attitudes were found in the tourism, hospitality and entertainment industries. There was a significant difference in attitude towards AI between employees at organizations with already implemented AI solutions (83% positive or very positive) and employees at organizations with no intention to implement them in the near future (35% positive or very positive). The current business attitude towards AI was very positive in organizations with already implemented AI solutions. Although the level of AI adoption differs among markets, AI solution implementation is definitely growing. The three main factors which had an impact on AI adoption in an organization were top management's attitude, competition and regulations. To overcome any negative impact from these factors, a strong relationship between the IT department and the management team must be established and continuously developed.

Among organizational factors the main role belongs to top management support. The findings confirmed the view expressed by many researchers (Sujata et al. (2019); Gregori and Holzmann (2020); Oosthuizen et al. (2020)) that only close collaboration between the business and IT teams at a company could ensure the smooth adoption and introduction of AI.

At the same time, the findings of the present study clearly show the high aspirations and expectations of companies for the utilization of AI, which completely matches the opinion expressed by other researchers (Bughin et al. (2018); Davenport and Ronanki (2018); Indriasari et al. (2019); Kaplan and Haenlein (2019a, 2019b); Pumplun et al. (2019)).

The correlation gained from the quantitative data conducted within the study showed a significantly high positive correlation between the top management support and department readiness factors (rs(336) = 0.74, $p < 0.001$). This confirms the theory that top management support influences department readiness for AI adoption and vice versa.

The present research discovered one unexpected outcome. Though according to the experts, trust in technological applications in general is another significant factor affecting an individual's attitude, we can conclude that trust is not always based on the knowledge and qualifications of the employees. There is a weak correlation between the knowledge and trust factors, and it may be related to experts' observation that most people have no idea how many AI-based applications they use on a daily basis. The authors believe that trust is more related to abstract/social than rational/technological factors. This aligns with the study conducted by Andrew (2017), which tested the correlation between employee attitudes and employee readiness for organizational change.

Regarding practical and managerial implications, the findings of this paper can assist top managers to keep in mind the challenges, advantages and risks that AI may entail for businesses and society. As AI technology is clearly related to IT, not all companies are ready to implement AI solutions because of low IT readiness. IT readiness can be measured not only by the level of IT infrastructure (Jöhnk et al. 2020), but also by the level of knowledge and data validity. Lack of knowledge in the IT department could be a factor influencing AI adoption. Firstly, many developers have a narrow scope of programming languages, but to develop a good AI system, one programming language is not enough. This raises the topic of education systems, which do not prepare system developers for work in the current digital era. To overcome these influencing factors, universities should adapt their courses in both business administration and IT programs. To strengthen collaboration between IT and management, technological universities should include lectures about business operations and IT's importance in them. Meanwhile, managers must understand the most basic AI principles to recognize when AI is fit for purpose.

The proposition based on the present study is as follows. Entrepreneurs must perform effective knowledge management on all organizational levels to successfully prepare the team for the digitalization era, while universities should adapt their courses according to the latest business and technological trends in both IT and business management programs to ensure stronger collaboration between these departments.

AI vendors must carefully study their prospective customers before targeting them directly, since, even if a customer is not interested in AI solutions, very likely they have a lack of knowledge about the broad use of AI applications.

## 5. Limitations and Future Research

The paper contributes to the very topical issue of introducing innovative technology in business and society while having a few limitations which are listed below.

Research was done in Latvia, and although Latvian society may be considered as very close to the other Baltic countries with regard to mentality and education level, the research results can be generalized only to some extent. In terms of restricting factors, the authors see limitations in the respondents selected, since they comprise persons only from Latvia. Therefore, a larger population should be selected in further research.

Another limitation relates to the research papers, as only research articles in English and indexed in the databases Web of Science and SCOPUS were analyzed. Given that research in the AI field is growing rapidly in such countries as Russia and China, some significant papers in other languages might have been omitted from the literature analysis.

The authors believe this study should be considered as the beginning phase for further investigation of issues related to AI and its utilization in business and public life.

As for further research, another survey should be implemented for longitudinal analysis and investigation of the attitudes towards innovative technology application adoption.

**Author Contributions:** Conceptualization, T.V. and I.L.; methodology, T.V. and I.L.; software, I.L.; validation, T.V. and I.K.; investigation, T.V. and I.L.; writing—original draft preparation, T.V. and I.L.; writing—review and editing, T.V. and I.K.; visualization, T.V. All authors have read and agreed to the published version of the manuscript.

**Funding:** This research received no external funding.

**Institutional Review Board Statement:** Not applicable.

**Informed Consent Statement:** Not applicable.

**Data Availability Statement:** Data were obtained during the research done in Latvia in 2020 as Master thesis research. Thesis with full set of data is publicly available in the RISEBA University library.

**Acknowledgments:** The authors would like to express gratitude to our respected experts and respondents who participated in the interviews and the survey. The results would not have been achieved without their passionate participation and input and valuable conversations and discussions during the interviews.

**Conflicts of Interest:** The authors declare no conflict of interest.

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
