# Peer review of "Artificial Intelligence: The Attitude of the Public and Representatives of Various Industries"

_jrfm, doi:10.3390/jrfm14080339_

Round 1

Reviewer 1 Report

The paper is interesting but it is necessary to strenghten the theoretical framework regarding the main topic, methods, discussion and conclusions.

The manuscript needs to be organized in a less schematic manner. The flow of assumptions, question research, theories, data and consideration should be more fluid and reasoned.

I suggest authors to reconsider also the style of the figures.

Results should be more properly explained and discussed. 

Discussion and conclusions should be more consistent and better explained.

Discussion and conclusions should be enriched with appropriate references.

Author Response

Dear reviewer,

Yours faithfully,

Tatjana

Reviewer 2 Report

Dear authors,

thank you for the opportunity to read your manuscript. I have several recommendations:

  1. Do not divide the introduction into chapters. Let it in one main chapter. Move the content as follows but without labels: Research Originality (add the reason for the research) and joint with Research Objective and Approach, finally, set Research Topicality and Contribution.
  2. Focus the chapter 2. only on Literature review.
  3. Focus the chapter 3. only on Methodology and Data
  4. Focus the chapter 4. only on Results
  5. Is Cronbach’s alpha coefficient above 0.6 enough? Why do not you prefer Cronbach’s alpha coefficient above 0.8? This value is recommended for social sciences.
  6. Rework the chapter Discussion. It is crucial part of the manuscript. Compare your empirical results to the results of similar study.
  7. Conclusions: provide short summary, set Limitations and Future Research in this chapter.

After reworking, I think the paper has its place in scientific journal JRFM.

I hope my comment will be useful for your future work.

Author Response

(The authors gave the same response as above.)

Reviewer 3 Report

The paper is interesting and deals vith some very important and relevant aspects of our lives and of the recent developments in the society. There are some aspects that need to be enhanced in order to increase the visibility and suitability of the paper.

Abstract

Please be more specific in the abstract by better highlighting the novelty of your research.  

Usually a paper start vith the research scope vhich is explained / detailed into several research objectives. Therefore I vould not recommend starting vith the research objective, but vith the research scope. 

You need also to state from the beginning the research gap and explain hov this gap is transposed into the research scope.  

I also do not recommend having that many subsection like 1.1., 1.2. etc. 

Vhy did you choose Latvia as a research context?

The literature reviev is nice developed, but papers from relevant journals vhich publish papers in AI are not cited. I recommend for instance to also search after relevant papers on the topic of AI in Computers in Human Behaviour. 

Research methodology. Nov... the funny thing is that you speak of 3  research questions ... but a paper should have 1 research question / research scope that should be further explained / detailed into research objectives. So you are kind of doing it  the other vay round. 

Hovever the aspects addressed in the research questions should have been also addressed in the literature reviev. 

"we decided to 172 extend the model and add social factors that influence the acceptance of AI in society" based on vhich criteria did you choose to do that?

The concepts from figure 1 should have been explained in detail in the literature reviev section. You should have also explained the hypothesis betveen the constructs.

2.2. and 3. Should be merge, as they both refer to the same thing

"The aim was to reach all social group" hov many are those? vhere can one find all these social groups? Are they different from one country to another?

"The first group was 193 considered to be AI technological experts, who have a solid knowledge of AI 194 technology," this seems to have been very subjective. Hov did you ensure that they are really experts in the field? Vhat criteria did you use? 

"The questionnaire was carried out through" vhere is the exact operationalisation of the cuestionnaire?

You need to shov some references for the choice of experts. Your approach does not seem to be a very rigurous one.

Cronbach alpa inder 0.7 is not a proper one. For the construct vith 0.6 ... it is not a reliable construct.

vhy does the number of respondents differ for the different constructs? This is really not proper!

Figure 6 is really different to figure 1. Vhy?

So vhich is your conceptual model`?

The research methodology is not very clear, nor is the presentation of results.

The paper has no discussions, meaning comparisons of ovn results vith previous findings of the literature.

Conclusions are veak. you need to enhance them by highlighting the theoretical and managerial contributions of the paper.

Reference list should be enhanced by citing more references dealing vith AI. 

Sorry for my spelling mistakes. The letter double v is not vorking on my laptop. 

Author Response

(The authors gave the same response as above.)

Round 2

Reviewer 2 Report

Dear authors,

thank you for your effort.

I recommend also including these articles related to artificial intelligence:

1. Durana, P., Perkins, N., and Valaskova, K. (2021). “Artificial Intelligence Data-driven Internet of Things Systems, Real-Time Advanced Analytics, and Cyber-Physical Production Networks in Sustainable Smart Manufacturing,” Economics, Management, and Financial Markets 16(1): 20–30. doi: 10.22381/emfm16120212.

2.  Lăzăroiu, G., Kliestik, T., and Novak, A. (2021). “Internet of Things Smart Devices, Industrial Artificial Intelligence, and Real-Time Sensor Networks in Sustainable Cyber-Physical Production Systems,” Journal of Self-Governance and Management Economics 9(1): 20–30. doi: 10.22381/jsme9120212.

Good luck in your scientific work.

Author Response

Dear reviewer,

in this amended version we have substantially enlarged the Chapter Conclusions and Discussions added supplementary data about the results of the research and reflected the scientific discissions on the results obtained.

You can briefly see the amended text as it is left in red color.

The text of the manuscript was proof-read by native English speaker, Dr. philology in English

The formatting of the manuscript according to the Journal requirements was done as well.

Reviewer 3 Report

The paper has been dramatically improved so it can be accepted. Congratulations.

Author Response

(The authors gave the same response as above.)
